



# Snow-vegetation-atmosphere interactions in alpine tundra

Norbert Pirk[1], Kristoffer Aalstad[1], Yeliz A. Yilmaz[1], Astrid Vatne[1], Andrea L. Popp[1,2], Peter Horvath[3], Anders Bryn[3], Ane Victoria Vollsnes[4], Sebastian Westermann[1], Terje Koren Berntsen[1], Frode Stordal[1], and Lena Merete Tallaksen[1]

[1]Department of Geosciences, University of Oslo, Oslo, Norway
[2]Hydrological Research Unit, Swedish Meteorological and Hydrological Institute (SMHI), Norrköping, Sweden
[3]Natural History Museum, University of Oslo, Oslo, Norway
[4]Department of Biosciences, University of Oslo, Oslo, Norway

**Correspondence:** Norbert Pirk (norbert.pirk@geo.uio.no)

**Abstract.** The interannual variability of snow cover in alpine areas is increasing, which may affect the tightly coupled cycles of carbon and water through snow-vegetation-atmosphere interactions across a range of spatio-temporal scales. To explore the role of snow cover for the land-atmosphere exchange of $CO_2$ and water vapor in alpine tundra ecosystems, we combined three years (2019-2021) of continuous eddy covariance flux measurements of net ecosystem exchange of $CO_2$ (NEE) and

evapotranspiration (ET) from the Finse site in alpine Norway (1210 m a.s.l.) with a ground-based ecosystem-type classification and satellite imagery from Sentinel-2, Landsat 8, and MODIS. While the snow conditions in 2019 and 2021 can be described as site-typical, 2020 features an extreme snow accumulation associated with a strong negative phase of the Scandinavian Pattern of the synoptic atmospheric circulation during spring. This extreme snow accumulation caused a one-month delay in melt-out date, which falls on the 92$^{nd}$-percentile in the distribution of yearly melt-out dates in the period 2001-2021. The melt-out dates

follow a consistent fine-scale spatial relationship with ecosystem types across years. Mountain and lichen heathlands melt out more heterogeneously than fens and flood plains, while late snowbeds melt out up to one month later than the other ecosystem types. While the summertime average Normalized Difference Vegetation Index (NDVI) was reduced considerably during the extreme snow year 2020, it reached the same maximum as in the other years for all but one the ecosystem type (late snowbeds), indicating that the delayed onset of vegetation growth is compensated to the same maximum productivity. Eddy covariance

estimates of NEE and ET are gap-filled separately for two wind sectors using a random forest regression model to account for complex and nonlinear ecohydrological interactions. While the two wind sectors differ markedly in vegetation composition and flux magnitudes, their flux response is controlled by the same drivers as estimated by the predictor importance of the random forest model as well as the high correlation of flux magnitudes (correlation coefficient $r = 0.92$ for NEE and $r = 0.89$ for ET) between both areas. The one-month delay of the start of the snow-free season in 2020 reduced the total annual ET by $50\%$

compared to 2019 and 2021, and reduced the growing season carbon assimilation to turn the ecosystem from a moderate annual carbon sink ($-31$ to $-6$ gC m$^{-2}$ yr$^{-1}$) to a source (34 to 20 gC m$^{-2}$ yr$^{-1}$). These results underpin the strong dependence of ecosystem structure and functioning on snow dynamics, whose anomalies can result in important ecological extreme events for alpine ecosystems.



## 1 Introduction

At northern latitudes, alpine tundra shares many similarities with arctic tundra regarding its appearance, dynamics, and role
in the Earth system. These ecosystems typically feature low vegetation, a shallow root zone with acidic soils, as well as a
complex pattern of inter-dependent plant, fungal, and microbial communities that emerge across a large range of spatial scales
(Walker et al., 2001). The vegetation is primarily limited by the supply of energy and nutrients, which is to a large degree
governed by the spatio-temporal variability of the snow cover. In alpine tundra, the surface energy balance, soil temperatures,

and nutrient availability are all directly affected by the presence of snow (Rixen et al., 2022). The snowpack moreover offers
plants protection against frost damage, dehydration, and mechanical damage from wind-blown snow particles in wintertime
(Mott et al., 2018). This protection comes at the price of longer-lasting snow cover limiting the growing season length for
plants (Vestergren, 1902), which is especially pronounced in topographic depressions where wind-blown snow accumulates.
The vegetation structure in these characteristic snowbed-ridge ecosystems will in turn influence the wind-blown snow transport

and thus modify the spatial variability of the snow distribution. These complex and consistent snow-vegetation interactions
give rise to repeating patterns in snow distributions (Sturm and Wagner, 2010) and are thus a key structuring process for alpine
tundra environments and an important control on land-atmosphere interactions (Odland and Munkejord, 2008).

Community ecologists have long recognized that plant associations form and thrive in specific ranges of environmental con-
ditions (Gleason, 1926; Whittaker, 1956). Responses to changes in site conditions depend on complex plant-plant interactions,

which can be highly context-dependent (Vandvik et al., 2020; Niittynen et al., 2020). Wipf et al. (2009) and Frei and Henry
(2022) analyzed plant phenology, growth, and reproduction in alpine and arctic shrubs, respectively, and found that reductions
in snow cover duration are beneficial for some, but not all tundra species. Niittynen et al. (2018), on the other hand, found a tip-
ping point at 20-30% decrease in snow cover duration at which accelerated species loss reduces the biodiversity in arctic-alpine
areas. Scharnagl et al. (2019) documented the expansion of shrubs in alpine ecosystems over a 40-year period, but argue that

plant community composition remained mostly intact, demonstrating a surprising resilience of alpine tundra plant communities
to ongoing global climate change. Similarly, Roos et al. (2022) show that experimental warming with ITEX chambers over
a 30-year period in alpine Norway only had a modest effect on the community composition, while nutrient additions caused
strong responses in vegetation dynamics.

The widespread greening of mountain slopes, as quantified by the Normalized Difference Vegetation Index (NDVI) (Jia

et al., 2003), can have profound impacts on the ecosystem's carbon and water balances through increased land-atmosphere
exchange of $CO_2$ and water vapor, i.e., the net ecosystem exchange of $CO_2$ (NEE) and evapotranspiration (ET). The link
between the carbon and water cycles in terrestrial ecosystems can be assessed through the ratio of NEE and ET, known as the
ecosystem water-use efficiency (as opposed to leaf-level water-use efficiency derived from photosynthesis and transpiration),
which provides another key indicator for ecosystem functioning under changing environmental conditions (Niu et al., 2011;

Schlesinger, 2020). In arctic tundra, NEE estimates show that longer growing seasons due to earlier snow melt-out may not
necessarily lead to stronger carbon assimilation, because tundra ecosystems may not be able to continue to take up $CO_2$ late
in the growing seasons (Groendahl et al., 2007; Zona et al., 2022). Evapotranspiration in high latitude ecosystems is normally





limited by net surface radiation and is thus typically small compared to total annual precipitation (Liljedahl et al., 2011), while lower latitude alpine grasslands can feature ET losses of more than 50% of total annual precipitation (Carrillo-Rojas

et al., 2019). Evapotranspiration has been found to decrease and feature higher interannual variability at higher elevations with sparser vegetation cover across a forest-shrub vegetation gradient in alpine Canada (Nicholls and Carey, 2021). While ET is currently a relatively small component in the water balance in arctic-alpine areas (Lackner et al., 2022), it is expected to increase considerably under climate change scenarios (Helbig et al., 2020), which makes it imperative to further constrain ET for ecological and hydrological models (Erlandsen et al., 2021).

Snow cover duration in the Northern Hemisphere is decreasing at an accelerating rate, even exceeding CMIP5 simulations (Derksen and Brown, 2012; Mudryk et al., 2020) with many arctic-alpine systems undergoing a transition from snow- to rain-dominated regimes (Bintanja and Andry, 2017; Arias et al., 2021). In Norway, increasing temperature and precipitation are associated with remarkably large reductions in snow cover duration both in historic estimates (Rizzi et al., 2018) and future projections (Hanssen-Bauer et al., 2017), with the exception of alpine areas which can even feature an increasing snow cover

duration. In addition to these mean climatic trends, there is accumulating evidence for increasing interannual variability of weather patterns as well as in the frequency and severity of extreme events (Easterling et al., 2000; Myers-Smith et al., 2020). While an increased frequency of extreme weather events can be expected to impact the land-atmosphere carbon exchange of otherwise undisturbed tundra ecosystems (Christensen et al., 2021), it has also been recognized that extreme weather events act as filters for leading edge species (Hampe and Petit, 2005) with, e.g., higher temperature demands. Thereby, extreme events

can provide a stabilizing mechanism in the mortality-recruitment balance of the ecosystem to prevent long-term vegetation shifts (Lloret et al., 2012; Beigaitė et al., 2022).

The joint response of NEE and ET to anomalies in snow cover duration in alpine environments, where snow-related extreme events may be expected to play the dominating role, is to our knowledge still understudied. The present study aims to explore the role of snow cover duration for ecosystem functioning in alpine tundra. We use the eddy covariance (EC) technique (Bal-

docchi, 2020) for near-continuous measurements of NEE and ET, allowing us to identify the key drivers of land-atmosphere interactions in the harsh meteorological conditions of alpine Norway. Complementary to direct EC flux measurements, we use high-resolution satellite remote sensing, in-situ ecosystem-type mapping of vegetation distributions, and long-term statistics of atmospheric circulation patterns to contextualize our findings. Finally, we argue that anomalies in snow cover duration constitute important ecological extreme events for the structure and functioning of alpine tundra ecosystem.

## 85  2   Materials and methods

### 2.1   Site description

The Finse site (Figure 1) is situated in an alpine valley (60.11°N, 7.53°E) at an elevation of 1210 m a.s.l. near the Finse Alpine Research Center in southern central Norway. The valley extends approximately along an east-west axis along which surface winds tend to be led by forced channeling (Whiteman and Doran, 1993). A large glacier, Hardangerjøkulen, is located

approximately 6 km south-west and the largest lake in the valley, Finsevatnet, lies 1 km west of the site. During summer, there



is a confluence of cool glacial meltwater and warmer non-glacial streams from the lake into the river Ustekveikja, which runs along the study site. During winter, the discharge reduces to base flow from groundwater inputs because the headwater rivers freeze over. The climate is arctic and features maritime influences with relatively mild winters (minimum 30-min average air temperature of $-30.4°C$ measured between 2019-2021) and cool summers (maximum 30-min average air temperature of $22.2°C$ measured between 2019-2021). The annual mean (1991–2020) air temperature is $-1.1°C$ with an average annual total precipitation of 967 mm. The site is likely largely permafrost-free, but more exposed areas with low snow depths in winter can feature isolated permafrost (Gisnås et al., 2014). The site features a low-alpine tundra ecosystem, dominated by lichen heathlands on wind-exposed ridges, as well as dwarf shrubs and mountain heathlands on the lee-sides. Willows dominated by Salix species form narrow flood plains along river margins. Snowbeds are common in wind-sheltered areas. In flat areas, water accumulates to form small wetlands and ponds.

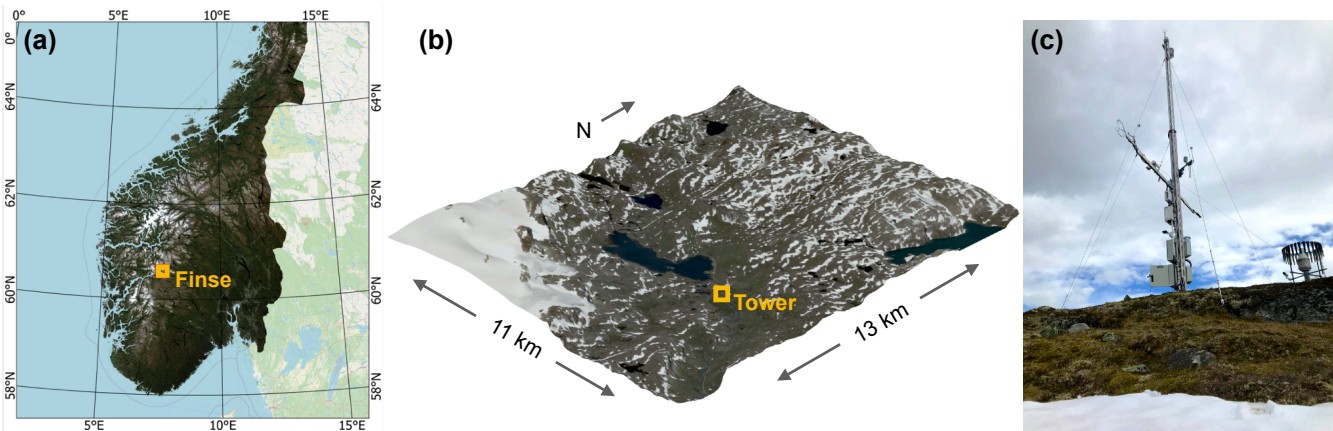

**Figure 1.** Location and environmental setting of the Finse flux tower. (a): Location of Finse in southern Norway (background data contributed by "Norge i Bilder" and Open Street Map). (b): Satellite image taken by Sentinel-2A on 31 Aug 2020 draped over an elevation model (DTM10 by Kartverket). (c): Image of the flux tower taken on 15 July 2020.

## 2.2 Flux measurements

Eddy covariance flux measurements of $CO_2$ and water vapor were established at the Finse site (registered as "NO-Fns" in FLUXNET (Baldocchi et al., 2001)) in 2016. Frequent technical problems with sensors and data loggers disrupted the first two years of operation, so the present study focuses on the period 2019-2021 with near-continuous flux data. The EC system consists of a CSAT3 three-dimensional sonic anemometer (Campbell Scientific, USA) and a Li-7200 closed-path infrared gas analyzer for $CO_2$ and $H_2O$ mixing ratios (Li-Cor, USA). Both instruments are installed on the northern end of a horizontal boom at $4.4$ m a.g.l. (Figure 1c) and sampled at a frequency of 20 Hz. The Li-7200 gas analyzer uses a 71 cm long heated intake tube (6 W) with a flow rate of $15$ L min$^{-1}$.





We processed the EC raw data to 30 minutes flux estimates following the conventional EC methodology (Gu et al., 2012)
using EddyPro version 6.2.0 (Li-Cor). We extract turbulent fluctuations from block averages, use an anemometer tilt correction
by double rotation, a constant time lag compensation, and a high- and low-pass filter correction following Moncrieff et al.
(2005) and Moncrieff et al. (1997), respectively. For quality control, we use statistical tests on the raw data proposed by
Vickers and Mahrt (1997) and the flagging system proposed by Foken and Wichura (1996) to filter out flux estimates that are
affected by instrument errors (e.g., rain or frost on the anemometer) or unfavorable micrometeorological conditions (e.g., lack
of stationarity or turbulent mixing at low wind speeds). Following Vickers and Mahrt (1997), we estimate the number of spikes,
drop-outs, as well as the absolute limits, amplitude resolution, skewness and kurtosis, and discontinuities for the pairs of raw
data time series involved in the respective covariance-based flux estimates and discard data exceeding the thresholds proposed
in the original paper. We also discard data with mean horizontal wind speeds below $1.5$ m s$^{-1}$, all fluxes with quality flag
2 in the scheme by Foken and Wichura (1996), as well as fluxes with quality flag 1 if they have relatively large magnitudes
(i.e., above $1.0$ $\mu$mol m$^{-2}$ s$^{-1}$ for NEE and $0.9$ mmol m$^{-2}$ s$^{-1}$ for ET). After filtering the flux time series for unfavorable
measurement conditions, we are left with $24\,076$ NEE and $22\,708$ ET valid half-hourly flux measurements, corresponding to
$46\%$ and $43\%$ coverage of the entire period from 2019-2021, respectively.

## 2.3 Ancillary measurements

There are a multitude of ancillary sensors on the Finse flux tower to quantify soil, surface, and atmospheric conditions during
our flux measurements. Near-surface air temperature ($T_{\mathrm{air}}$) is measured by a resistance temperature detector (PT-100) mounted
in a radiation shield at 2 m a.g.l. Growing degree days (GDD) is calculated from $T_{\mathrm{air}}$ according to its standard definition using
a base temperature of $0°$C. Vapor pressure deficit (VPD) is derived from measurements of $T_{\mathrm{air}}$ and relative humidity (HMP155,
Vaisala, Finland) mounted at 2 m a.g.l. Soil temperature ($T_{\mathrm{soil}}$) and soil volumetric water content (VWC) are measured at a
depth of 8 cm (CS650, Campbell, USA). For incoming shortwave and longwave radiation (SW$_{\mathrm{in}}$ and LW$_{\mathrm{in}}$, respectively) we
use a ventilated and heated radiometer (CNR4, Kipp&Zonen, Netherlands) mounted on a south-pointing boom at 4 m a.g.l. The
same sensor is used to measure surface broadband albedo. Snow depth (SD) is measured by a laser distance sensor (SHM30,
Jenoptik, Germany) at a location about 4 m north-east of the tower. Skin surface temperature ($T_{\mathrm{surf}}$) is measured with an
infrared radiometer (SI-411, Apogee, USA) at approximately the same location as the snow depth measurement. All these
ancillary sensors are sampled every 10 s, filtered for corrupted measurements, and aggregated to 30 minute average values. Due
to data logger problems, approximately $2\%$ of the 30 minute intervals lack valid local measurements in the period from 2019-
2021. For atmospheric and surface variables these short gaps are filled with estimates derived from a simple linear regression of
the respective variable against its corresponding estimate from ERA5 atmospheric reanalysis data (Hersbach et al., 2020). Soil
variables, which vary on longer timescales, are filled with a linear interpolation of neighboring measurements. The resulting
time series are shown in Figure S1 in the Supplement.



## 2.4 Flux gap-filling

As gaps in the EC flux time series can occur systematically depending on environmental conditions, we use gap-filling to avoid biases in our annual flux budgets. To allow for a complex range of biogeochemical interactions, we developed a random forest regression model (Breiman, 2001; Kim et al., 2020) of the fluxes with 12 predictors quantifying the environmental conditions (Figure S1 in the Supplement). Ten of these predictors are measured directly at the flux tower, as described in Section 2.3, while two additional ones—the fractional snow-covered area (FSCA) and Normalized Difference Vegetation Index (NDVI)—are derived from remote sensing (Section 2.6). These 12 predictors are chosen to provide a robust and detailed characterization of each 30-minute flux period, including soil, surface, and atmospheric conditions. Some of the predictors are correlated, at least in some parts of the predictor space, which must be considered when interpreting the predictor importance obtained from the random forest regression model (Gregorutti et al., 2017). Since even highly correlated predictor pairs may capture nuances of potentially important flux dynamics, we chose to use their information in our gap-filling routine despite the partial redundancy (see discussion in Section 4.2). Note that the resulting flux estimates from the random forest regression model are only used to fill gaps in the flux time series, i.e., not to replace valid flux measurements.

The setting of the landscape at the Finse flux tower favors a bi-modal distribution of wind directions along an east-west axis. To account for the potentially different surface cover of footprint East and West, we split the dataset into the two main wind directions (wind directions above and below $180°$) and gap-filled these subsets separately. Such splits are common to prevent annual flux budgets to depend on the distribution of the wind directions (Griebel et al., 2016).

We use the random forest regression implementation provided by the Scikit-learn Python module (Pedregosa et al., 2011), with 2000 trees per forest and otherwise default parameters. The random forest regression model is trained on valid EC flux measurements that have passed quality control. To assess potential overfitting that would limit the generalization of the model to unobserved data, we also trained separate models using only 80% of our valid dataset, keeping 20% for testing through independent evaluation (validation). The coefficients of determination of these random forest models ranged between $0.85 \leq R^2 \leq 0.95$ across the two flux types (NEE and ET) and wind sectors (Table S2 in the Supplement). We also tested reducing model complexity by limiting the number of predictors that each tree is randomly assigned or by reducing the maximally allowed depth of each tree. However, the resulting evaluation statistics indicated that overfitting is not a problem for our case with more than 1000-times more data points than predictors.

## 2.5 Footprint characterization by ecosystem types

The footprint function of each valid 30-minute flux is estimated based on friction velocity ($u_*$), wind direction, Obukhov length ($L$), and cross-wind variance ($v_{var}$) (all from EC measurements), as well as boundary layer height (linearly interpolated estimates from single-level ERA5 hourly data), and a tundra-typical roughness length of 1 cm, following the flux footprint model by Kljun et al. (2015). The resulting $1 \times 1$ m$^2$ resolution flux-weight maps are clustered by ecosystem type using a map of the area created in-situ by Bryn and Horvath (2020) at a mapping scale of $1:5\,000$ to assess the flux contributions of different surfaces. The implemented Nature in Norway (NIN) hierarchical mapping system (Halvorsen et al., 2020) consists





of a total of 741 minor and 92 major ecosystem types. Of these, 43 minor and 13 major types are found in the study area at Finse. For the purpose of this study, the ecosystem types are further reclassified into seven main type categories (Table S1).

Fens comprise all open mires with peat-dominated ground layers that, in addition to rainwater, are also fed by groundwater that has been in contact with the mineral soil. Flood plains include open alluvial sediments regulated by balancing sedimentation and erosion. Mountain heathlands are characterized as naturally open ecosystems above the climatic forest limit, dominated by dwarf shrubs (*Empetrum nigrum*, *Salix* spp. and ericaceous species), herbs, graminoids and bryophytes. Exposed ridges (and lichen heathlands) are confined to convex terrain and areas that lack permanent snow cover in winter during periods with

extremely low temperatures, freeze-drying conditions, and physical wind abrasion, dominated by specialized stress-tolerant lichens, mosses and vascular plants. Moss-dominated snowbeds are characterized by a combination of shortened growing seasons due to prolonged snow cover and shelter of the vegetation against low temperatures and wind abrasion during winter. Moderate snowbeds occupy the lower lee-sides of the topographical "ridge-snowbed gradient" in alpine and arctic areas, while late and extreme snowbeds can be found at the lowest part of the topographical depressions where snow does not completely

melt each year.

### 2.6 Satellite remote sensing and synoptic patterns

The fractional snow covered area (FSCA) and Normalized Difference Vegetation Index (NDVI) are retrieved from multispectral satellite imagery covering the $3 \times 3\,\mathrm{km}^2$ area around the Finse flux tower at a ground sampling distance of 10 m. We use surface reflectances (i.e., level 2 products) from the Sentinel-2 and Landsat 8 satellites in the following 6 wavelength bands: blue

($\simeq 0.49\,\mu$m), green ($\simeq 0.55\,\mu$m), red ($\simeq 0.65\,\mu$m), near-infrared ($\simeq 0.85\,\mu$m), shortwave infrared 1 ($\simeq 1.6\,\mu$m), and shortwave infrared 2 ($\simeq 2.1\,\mu$m). The data were obtained from Google Earth Engine (Gorelick et al., 2017), which is a cloud-based platform that harvests these open datasets from the original data sources, namely Copernicus (Sentinel-2) and USGS (Landsat 8). The FSCA is retrieved using the spectral unmixing approach described in Aalstad et al. (2020). The NDVI, which is commonly used as a proxy for surface greenness, leaf area, and vegetation development, is calculated according to its standard

definition (Jia et al., 2003). To avoid artifacts in the satellite-based surface reflectance data that can occur due to clouds, we manually selected cloud-free scenes. This selection provided a total of 93 Sentinel-2 scenes and 20 Landsat 8 scenes for the entire study period, resulting in an average of around four cloud-free scenes per month. The stack of cloud-free retrievals of FSCA and NDVI were interpolated in time, independently for each pixel using Gaussian process regression (Rasmussen and Williams, 2005) with an exponential kernel and automatic relevance detection.

For long-term snow cover statistics, we use daily Normalized Difference Snow Index estimates from the MODerate resolution Imaging Spectroradiometer (MODIS) at 500 m spatial resolution to retrieve the FSCA based on a linear relationship (Salomonson and Appel, 2006) for all water years (September-August) from 2001 to 2021. MODIS is an optical satellite-based sensor currently onboard two polar-orbiting satellites, namely Terra and Aqua, that observe the Earth following the same orbit tracks approximately three hours apart. These measurements from the MODIS sensors include gaps mainly due to cloud cover.

By merging two MODIS-based snow products from Terra (MOD10A1; Hall et al., 2015a) and Aqua (MYD10A1; Hall et al., 2015b), we are reducing these gaps for a given day. Subsequently, a temporal cloud gap-filling algorithm following Hall et al.





(2019) is applied to this merged product to obtain gap-free daily FSCA estimates. For each pixel, snow melt-out dates are determined as the last day in a water year with FSCA greater than $0.25$ during a period with at least five consecutive snow cover days. We averaged these estimates from the four closest MODIS pixels to the Finse tower to determine the snow melt-out dates

at our site. This snow cover estimation from the MODIS data agrees qualitatively with a visual inspection of daily webcam imagery available at the Finse research station (www.finse.uio.no/news/webcam/). We also evaluated the snow cover duration from the MODIS using the higher resolution (Sentinel-2 and Landsat 8) retrievals as a reference during an overlap period (from 2017 to 2021) and found a close agreement with an root mean square error of 6 days and a correlation coefficient $r = 0.98$ for the 9 km$^2$ study area.

To estimate the exceedance probability of the late snow melt-out in 2020, and to identify whether or not this year was an extreme year, we fit a beta distribution—a commonly used distribution with two shape parameters ($\alpha$ and $\beta$) for double-bounded random variables—to the melt-out dates from the MODIS dataset with the maximum likelihood method. Using gamma, logit-normal, and Generalized Extreme Value distributions for the fit to the melt-out dates only has minimal influence on the resulting exceedance probability.

To characterize the atmospheric circulation pattern during the heavy snowfall in 2020 compared to other years, we analyze the correlation between wintertime precipitation and the Scandinavian Pattern Index (SCA, referred to as Eurasia-1 pattern by Barnston and Livezey (1987)) estimated from monthly ERA5 data using the KNMI Climate Explorer (Trouet and Van Old-enborgh, 2013). Similar to the North Atlantic Oscillation Index, the Scandinavian Pattern Index is based on the surface air pressure difference between the subtropical and subpolar regions. We focus this analysis on February as the central winter

month, which typically features large snowfall events.

## 3 Results

### 3.1 Snow-vegetation interactions at Finse

Figure 2a shows the results of the spatial clustering of remotely sensed melt-out dates from the combined Sentinel-2 and Landsat 8 retrievals to the ecosystem type map. Different ecosystem types are associated with markedly different snow melt-out

characteristics: While fens and floodplains melt out relatively simultaneously (i.e., with small within-ecosystem type variance), mountain and lichen heathlands melt out slightly more variably. Moderate and late snowbeds melt out most variably and on average one to two months after the other ecosystem types. Figures 2a-d show that all three years exhibit similar relative melt-out patterns, but 2020 had considerably later overall melt-out dates. This difference can largely be attributed to differences in snow accumulation in winter and spring, exemplified by a maximum snow depth of 205 cm in 2020, compared to 52 cm in

2019 and 70 cm in 2021 as measured at the flux tower (Figure S1 in the Supplement).

The relative response of the vegetation development to the snow cover is assessed by our NDVI estimates. Figures 2e and i show that fens and mountain heathlands have the highest and similar mean and maximum NDVI, followed by flood plains, lichen heathlands and moderate snow beds (with similar NDVI statistics), and finally late snow beds. Averaged across the three summer months, NDVI was lower in 2020 compared to 2019 and 2021 (Figures 2e-h). The maximum NDVI of each ecosystem



type, however, was very similar in all years (Figures 2h and i). This robustness of annual-maximum NDVI indicates that while the summertime-average leaf area and greenness was reduced in the snow-rich year of 2020, the peak NDVI still reached the same level as in the other years. The only exception are late snowbeds (and to a smaller degree moderate snowbeds), where the extreme snow accumulation of 2020 noticeably inhibited vegetation growth. The effect of snow melt-out date on annual-maximum NDVI can also be seen from the spatial anti-correlation of these variables in the 9 km$^2$ satellite area, which was $r = -0.43$ on average and especially strong in 2020 ($r = -0.36$ in 2019, $r = -0.52$ in 2020, and $r = -0.40$ in 2021).

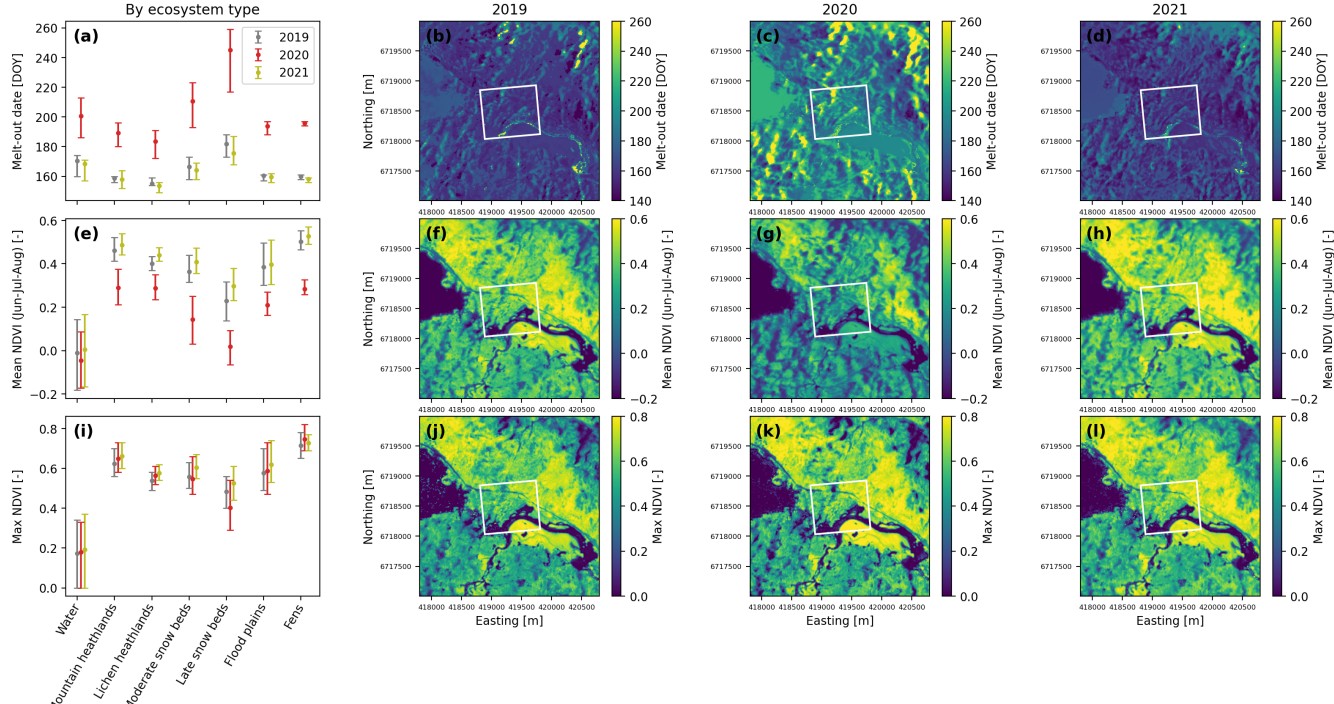

**Figure 2.** Remotely sensed melt-out dates and NDVI statistics from the combined Sentinel-2 and Landsat 8 retrievals for 2019, 2020, and 2021. Left: Clustered into different ecosystem types. Dots represent mean values and error bars the inter-quartile ranges of the distributions within each ecosystem type. Middle and right: Maps for the 9 km$^2$ area around the flux station. The white rectangle represents the area around the flux tower where ecosystem-type mapping and clustering was performed (same as in Figure 4a).

Most of the snowpack at Finse built up in only a few major precipitation events and almost half of the maximum snow depth accumulated during two snowfall events in the winter months (Figure S1). The intensity of wintertime precipitation in southern Norway can in part be explained by the synoptic atmospheric circulation pattern as exemplified by the large anti-correlation between the Scandinavian Pattern Index and February precipitation shown in Figure 3a. This correlation map shows a strong north-south gradient across Europe with large absolute correlations in many areas, indication a strong association between the SCA index and precipitation in February. In winter 2020, the Scandinavian Pattern Index for February exhibited its lowest value in the ERA5 record (1950-2021), while 2019 and 2021 were close to the mean value (Figure 3b). The associated large snowfall





events in the winter of 2020 contributed to the extremely late snow melt-out (Figure 3c). The beta distribution fitted to the

melt-out dates (maximum likelihood shape parameters $\alpha = 3.42$ and $\beta = 18.98$) shows that 2020 falls on the $92^{nd}$-percentile

of the distribution, rendering 2020 an extremely snow-rich year. The snow melt-out date in 2020 ranked $2^{nd}$ in this time series

of 21 years (only exceeded in 2015).

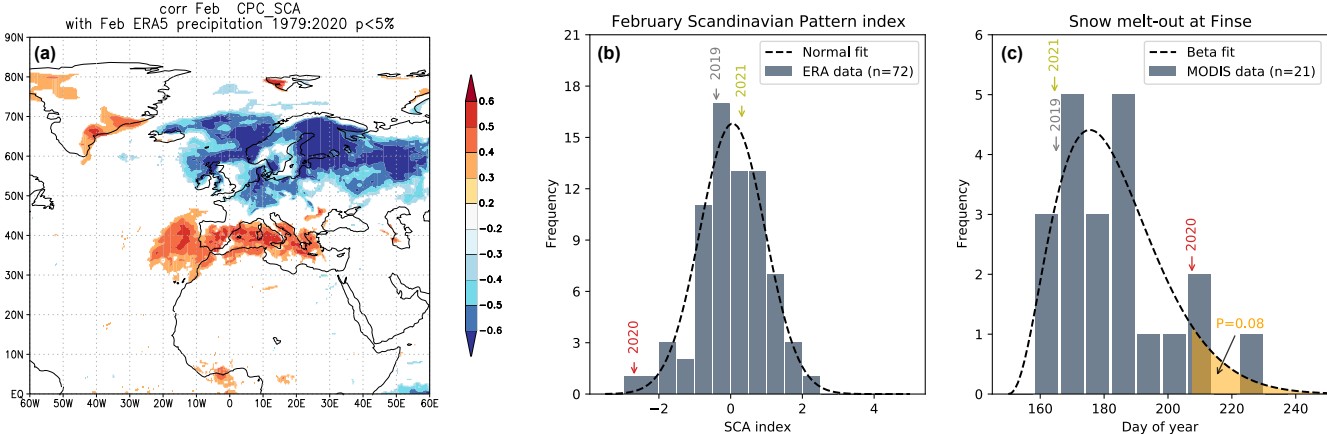

**Figure 3.** Synoptic atmospheric circulation patterns and snow cover statistics. (a): Correlation map between Scandinavian Pattern Index (SCA) and precipitation in February. (b): Histogram of all February values of the Scandinavian Pattern Index in the ERA5 period (1950-2021) along with the fitted normal distribution. (c): Histogram of the snow melt-out day at Finse derived from MODIS data (2001-2021) along with the fitted maximum likelihood beta distribution.

### 3.2 Flux dynamics in the two footprints

Figure 4 provides a characterization of the footprint of the EC flux measurements during the period 2019-2021 in terms of the weighted contribution of each ecosystem type. The footprint of each flux averaging period (30 minutes), as well as the

average footprint of the entire flux time series, receives contributions from a broad mixture of ecosystem types. Overall, snowbed surfaces have a footprint-weighted contribution of 30% to the total footprint area, followed by heathlands (29%), water surfaces (25%), as well as fens and flood plains (14% together) (Table S1 of the Supplement). The bi-modal distribution of wind directions seen in Figure 4b aligns along the east-west direction of the valley, facilitating the binary separation between the easterly and the westerly footprint in the flux analysis. Footprint East is characterized by a larger fraction of water surfaces

and late snowbeds, while footprint West has a larger fraction of fens and moderate snowbeds, with a denser vegetation cover.

Net ecosystem exchange of footprint West (Figure 5a) shows both diurnal and annual cycles as may be expected for a summer active, high-latitude tundra ecosystem. The winter and spring seasons are characterized by small, but steady $CO_2$ releases of about 0.1 $\mu$mol m$^{-2}$ s$^{-1}$. The start of the growing season with daily average $CO_2$ uptake fluxes occurs two weeks after the estimated day of snow-melt. Summer nights show the largest $CO_2$ releases, but daytime $CO_2$ uptake during summer

exceeds these nighttime releases in magnitude by a factor of about five. These flux dynamics are similar across the years, with





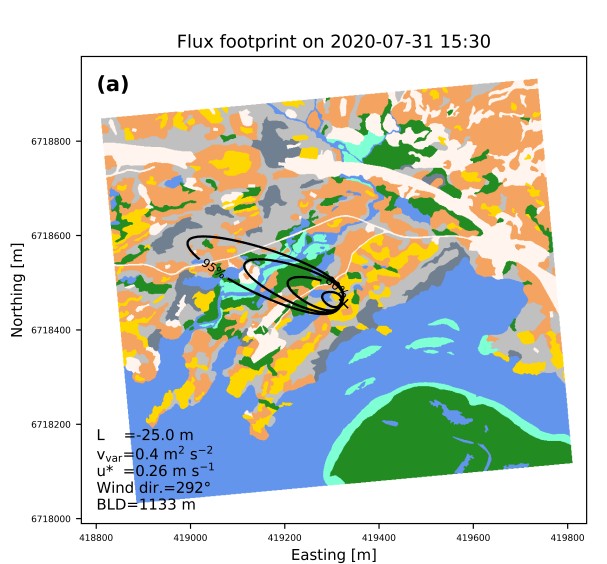

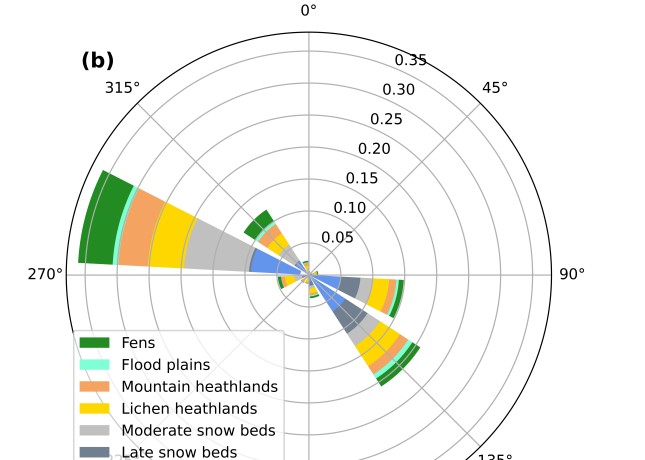

**Figure 4.** Flux footprint characterization. (a): Example of a footprint estimate for one flux averaging period during daytime summer conditions with unstable stratification. The background map shows the seven main ecosystem type categories, while the black lines show the contours of the 50%, 80%, 90%, and 95% cumulative footprints (from inside to outside), respectively. (b): Integrated footprint function averaged over all valid NEE measurements during 2019-2021 and plotted by the corresponding wind sectors. Colors indicate the footprint-weighted contribution of each ecosystem type. The weights sum up to about $0.91$ corresponding to the fraction of the EC footprint that is classified by the ecosystem type map.

the important difference of a much shorter growing season in 2020. The net ecosystem exchange of footprint East (not shown) exhibits the same dynamics as footprint West, with a correlation coefficient of $r = 0.92$ between the two, albeit with a smaller flux magnitude (slope of linear regression of $0.61$). The predictor importance of the random forest analysis (Figure 5b) indicate that shortwave incoming radiation, surface temperature, and NDVI are the main controls on NEE in both footprints.

Evapotranspiration shown for footprint West in Figure 5c also highlights different dynamics during the growing season compared to the rest of the year. During winter and spring, as well as during summer nights, small evaporation, sublimation, and condensation fluxes occur interchangeably, depending on the physical state of the ground and the surface layer air. Spring 2020 had a period where condensation dominated daytime fluxes, but compared to summertime ET these fluxes are relatively small. For daytime periods during the growing season, fluxes can be an order of magnitude larger and hence dominate the 280    annual ET budget. As seen for NEE, ET dynamics are similar across the years, but with reduced fluxes in the growing season of 2020. In footprint East, ET largely follows the same dynamics as in footprint West (with a correlation coefficient of $r = 0.89$) but with a smaller flux magnitude (slope of linear regression of $0.58$). The predictor importance for ET (Figure 5d) indicated that vapor pressure deficit, surface temperature, and incoming shortwave radiation act as the most important drivers for both footprints.



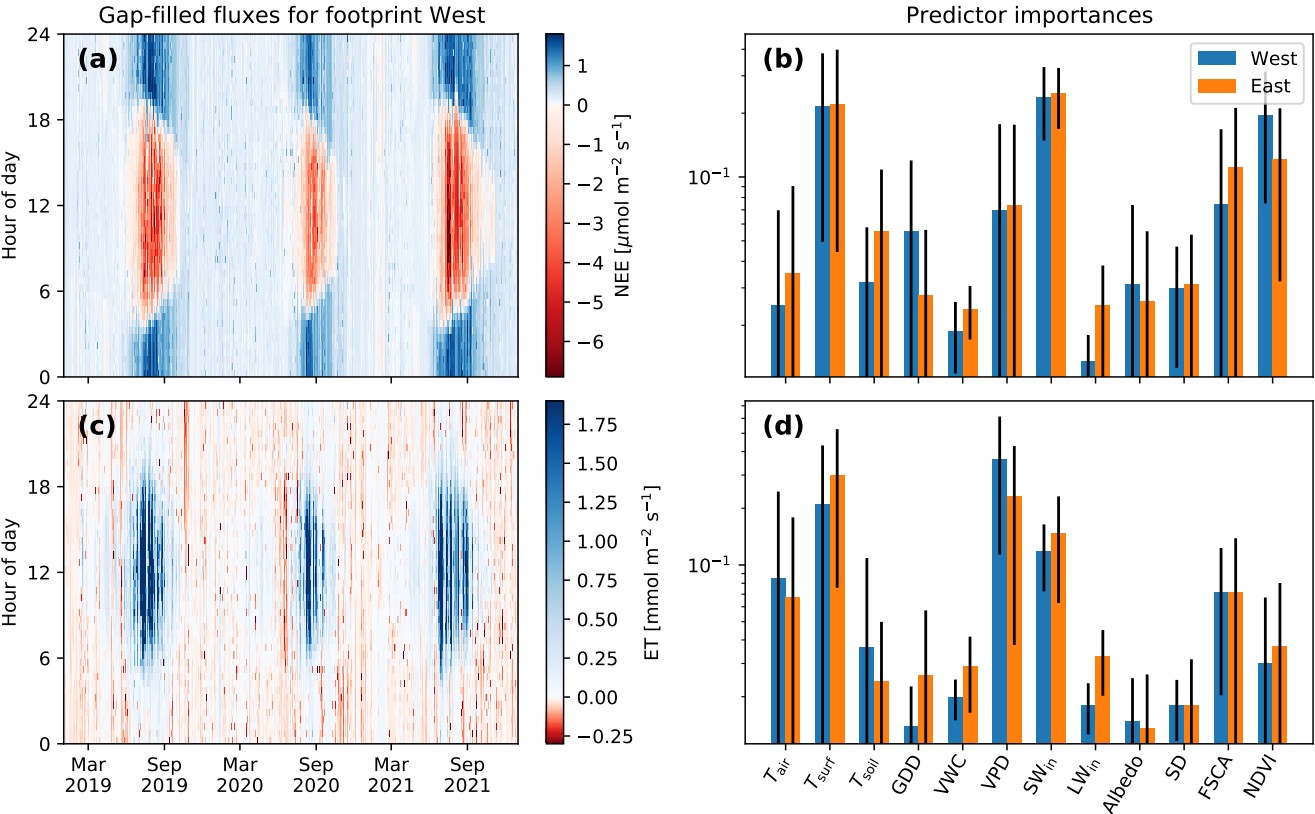

**Figure 5.** Flux dynamics and drivers. Left: Gap-filled NEE and ET as fingerprint plots for footprint West. Right: Predictor importance of the random forest regression models of both footprint East and West. Black error bars indicate the standard deviation across the 2000 trees in the respective random forests.

## 3.3 NEE and ET budgets

Figure 6 shows the cumulative NEE and ET of both footprints for each year. For NEE, more than nine months of the year typically feature small emissions of $CO_2$. In 2019 and 2021, these emissions were exceeded by an intense $CO_2$ uptake during the growing season, rendering the site a moderate annual carbon sink of between $-6$ and $-31$ gC m$^{-2}$ yr$^{-1}$ in both footprints (see Table S3 in the Supplement for detailed growing season statistics). In the extreme snow year 2020, however, the growing season $CO_2$ assimilation was too small to offset wintertime emissions, causing the annual carbon balance to become positive (carbon source) in both footprints (between 20 and 34 gC m$^{-2}$ yr$^{-1}$). For footprint West, the effect of a shortened growing season even made the ecosystem a stronger annual carbon source than it was a sink in the other two years. The large snow volumes in 2020 not only delayed the start of the growing season, but also markedly prevented ground freezing in wintertime (Figure S1 in the Supplement). The resulting higher soil temperatures allow for increased microbial activity in the soil, which can explain the slightly larger wintertime $CO_2$ emissions observed in 2020. Summer 2021 is characterized by larger $CO_2$



uptake fluxes compared to 2019, which may be caused by slightly higher GDD resulting in slightly higher NDVI values in 2021 (Figure S1 in the Supplement). The end of the growing season, when the ecosystem returns to being a net daily $CO_2$ source, occurred between September 5th and September 26th, and is associated with a fall in NDVI.

300 The ET loss is largely dominated by growing season fluxes. The annual ET loss is larger in the more densely vegetated footprint West, but is still less than 10% of total annual precipitation in both footprints. The ET contribution to the water balance is particularly low in 2020, when the short growing season resulted in a decrease in annual ET of almost 50%. Collectively, these observations are consistent with the notion that a large portion of ET stems from transpiration through stomatal fluxes, as opposed to non-stomatal evaporation from bare ground or interception storage.

305 The carbon uptake and ET loss integrated over the growing season can be used to estimate the integrated water-use efficiency (WUE) of the ecosystem, also termed WUE of productivity. This comparison shows that while both footprint areas are characterized by similar WUE in 2019 and the extreme snow year 2020 (range between 1.20 and 1.37 $\mu$mol-$CO_2$ mmol-$H_2O^{-1}$), the following year of 2021 showed a marked increase in WUE to values between 1.77 and 1.96 $\mu$mol-$CO_2$ mmol-$H_2O^{-1}$ (Table S3 in the Supplement). On average, these differences correspond to an increase in ecosystem water-use efficiency of about 47%.

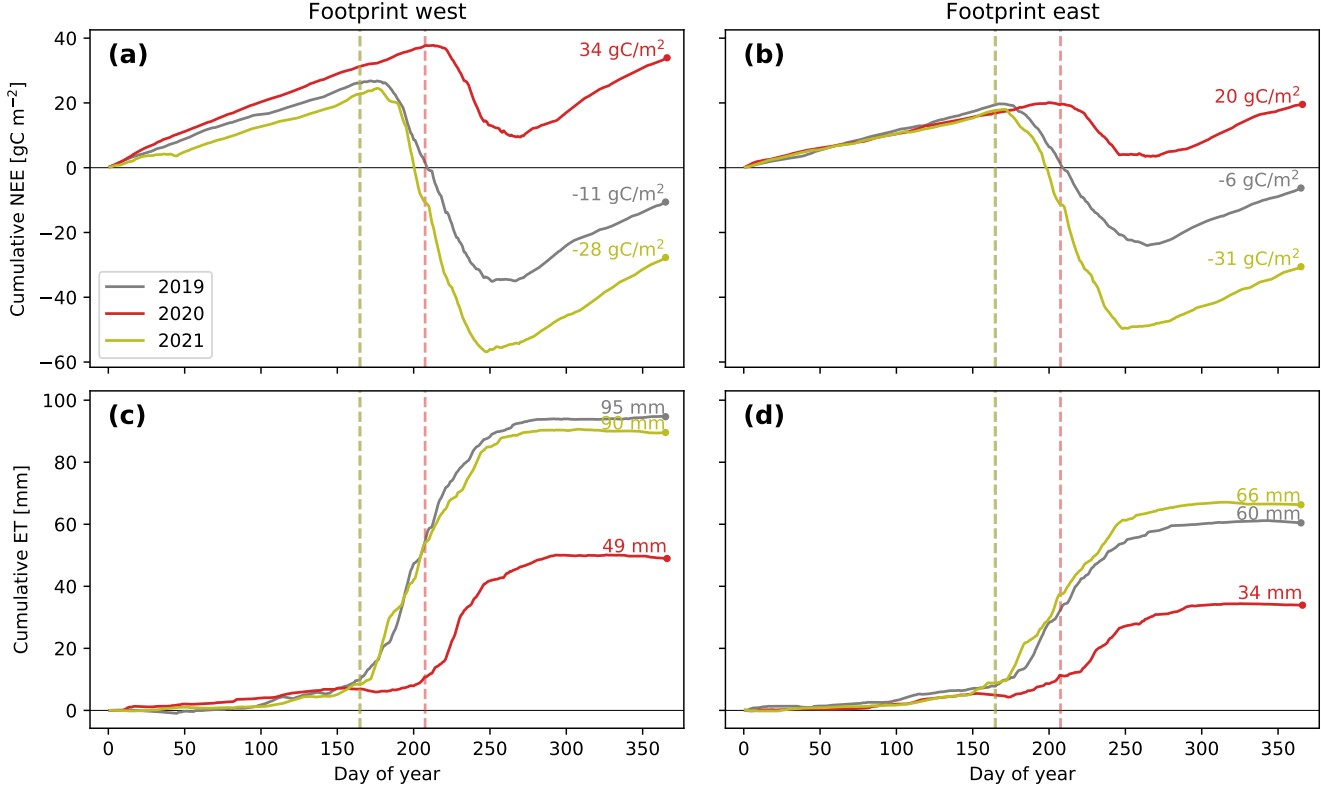

**Figure 6.** Cumulative sums of NEE (top) and ET (bottom) for different footprint areas (columns) and years (lines). Stipulated lines indicate the snow melt-out date estimated from MODIS (which was equal in 2019 and 2021).





## 4 Discussion

### 4.1 Snow cover anomalies as extreme events in alpine ecosystems

The ecology of alpine tundra along the Scandinavian mountain chain is strongly regulated by the spatial distribution and longevity of the snow cover (Dahl, 1956; Odland and Munkejord, 2008; Niittynen and Luoto, 2018), which is confirmed by the persistent relation between snow melt-out date and the ecosystem type in the present study (Figure 2a). Compared to the boreal region, alpine tundra in Norway is less affected by disturbances from wildfires (Jolly et al., 2015), heat waves (Ciais et al., 2005), or insect outbreaks (Heliasz et al., 2011). Although there is a history of free-ranging domestic animals (Ross et al., 2016) and reindeer (Kolari et al., 2019), as well as cycles of rodent population outbreaks (Kausrud et al., 2008), snow cover anomalies are likely driving the most consequential structural shifts of this ecosystem's functioning. The identified link between snowfall intensity and the Scandinavian Pattern index of the synoptic atmospheric circulation gives a new perspectives for our understanding of snow cover anomalies at Finse. The associated hydrological and biochemical consequences in areas with delayed snow melt-out can include enhanced groundwater recharge and subsequent discharge (Jasechko et al., 2014), wetter soils, higher pH, and more nutrients in the soils (Moriana-Armendariz et al., 2022). Our flux measurements clearly indicate the direct reduction of NEE and ET in extreme snow years. Perhaps somewhat surprisingly, our NDVI analysis reveals that the tundra vegetation still developed the same leaf area and greenness in the extreme snow year of 2020 compared to the other two years. In areas with short growing seasons such as at Finse, a quick development of new leaves on deciduous shrubs is needed to reduce the risk of missing the opportunity to grow. This quick leaf development is accomplished through mobilization of below-ground stored assimilates from the previous year (Karlsson, 1985; Körner and Renhardt, 1987; Tonjer et al., 2021). Although beyond the scope of this study, such below-ground responses could be explored using factorial experiments with minirhizotron tubes in climate chambers (Blume-Werry et al., 2019). Further, in order to maximize NDVI, leaves that are pre-planned in buds must not be injured by low winter temperatures or frost spells after bud break (Wipf et al., 2009). In our study, the late melt-out in 2020 came after a good growing season in 2019, probably securing assimilate storage and well developed buds. The shoots of 2020 came out late, thus probably avoiding frost spells. Accordingly, NDVI reached a high maximum level although the remainder of the growing season was too short to secure a negative NEE that year. So if extreme snow accumulations become more frequent, the observed reduction of growing season carbon assimilation could provide a mechanism for a trajectory for expanding snowbeds and reduced vascular plant cover in some ecosystem types in alpine regions, which would slow down the widely observed advances of birch trees to alpine regions and the shrubification of tundra areas.

In the growing season 2021, our flux measurements show a marked increase in WUE of the ecosystems in both footprint areas. Compared to 2019, NDVI, GDD and NEE were higher in 2021, whereas ET stayed approximately the same (see Figure S1 and Table S3 in the Supplement). Since our measurements do not suggest a clear mechanism for this increase in WUE, it remains an open question if it is due to a carry-over effect of the previous snow-rich year or the direct response to slightly higher air temperature (as suggested by Wang et al. (2020)) or lower volumetric water content in the soil due to less frequent precipitation (Figure S1 in the Supplement).





The Finse area typically features a several-months long ablation season with patchy snow cover (Aas et al., 2017). The main drivers of melt-out variability are likely differences in wind exposition and surface orientation with respect to prevailing wind directions and insolation. Likewise wind-driven snow melt by heterogeneous sensible heat fluxes can play a decisive role for

snowmelt dynamics (van der Valk et al., 2022). While our partitioning of snow patchiness into within- and between-ecosystem type variability (Figure 2a) comes with some sensitivity to our choice of merging the minor ecosystem types, it may offer new insights into the drivers of snow melt and discharge generation on the landscape scale. In this context, it would also be interesting to investigate the thermal impact of (relatively warm) groundwater upwelling (e.g., in streams or in the fens) on snowmelt and associated changes such as soil moisture and NEE. A companion paper to the present work assesses how

changing snow patterns affect the hydrology and aquatic biogeochemical cycling including greenhouse gas dynamics in the Finse area.

## 4.2 Eddy covariance measurements in mountainous environments

Mountainous topography can create characteristic boundary layer flow features due to, e.g., orographic gravity waves and cross-valley circulation cells that add effective transport mechanisms to the otherwise isotropic turbulent mixing (Lehner and

Rotach, 2018; Adler et al., 2021). At Finse, the nearby Hardangerjøkulen Glacier is likely to induce a secondary circulation pattern due to strong surface temperature gradients, especially during summertime. Such mesoscale disturbances can add noise and even systematic biases to EC flux estimates since they are not accounted for in the fundamental equation of EC (Gu et al., 2012). One manifestation of such problems can be seen in the non-closure of the surface energy budget that is typically of the order of 20% (Wilson et al., 2002) and increasing with landscape heterogeneity (Stoy et al., 2013). Ramtvedt and Pirk

(2022) showed that net radiation varies significantly over different surfaces around the Finse flux tower. However, even after correcting for these differences, the energy imbalance at Finse was still about 50%—similar to values found at other alpine sites (Rotach et al., 2004). Vertical temperature gradient measurements at our tower (Figure S2 in the Supplement) indicate stable stratification for about 83% of the period 2019-2021, with temperature gradients between the 2 m and 10 m levels as high as 6°C during low-wind conditions. The resulting surface-based temperature inversions and stable boundary layer flows are

known to feature flux divergences that can in part explain the observed energy imbalances (Mahrt et al., 2018). It is, however, still unclear to what extent this problem affects other flux estimates such as NEE. Adapted data processing techniques (Sievers et al., 2015) or alternative measurements using fiber-optics (Fritz et al., 2021) or drones (Pirk et al., 2022) could be employed to further investigate the surface energy imbalance at Finse, ideally in combination with other mountainous sites.

## 4.3 Statistical flux gap-filling

The random forest regression model for flux gap-filling performed well for our dataset, yielding generally low root mean square errors and high values of the coefficient of determination ($0.85 \leq R^2 \leq 0.95$ in test datasets, see Table S2 in the Supplement). Unlike other commonly used gap-filling algorithms (Reichstein et al., 2005), the random forest model assumes no functional relationships between drivers and fluxes, and allows for a complex statistical representation of biogeo-chemical and -physical interactions that give rise to fluxes. The disadvantage is that random forest models cannot extrapolate to regions of the input-

output space that are outside the observed range. For near-continuous datasets, such as our NEE and ET fluxes, this is not a critical limitation, but datasets with larger gaps during extreme environmental conditions would be more challenging to gap-fill with random forest models. While the predictor importance hints towards the underlying processes and can help develop new hypotheses about biogeo-chemical and -physical interactions, their interpretation is complicated by correlations among the predictors, as is common for statistical and machine learning models with multiple predictors (Gregorutti et al., 2017). The

predictor importance in our flux model is in good agreement with our prior expectations about the flux drivers. At the same time, our analysis emphasizes the need for high-quality ancillary measurements nearby flux towers. In accordance with the No Free Lunch theorems of optimization (Wolpert and Macready, 1997), a range of different statistical and machine learning models would likely fit the data equally well. Bayesian additive regression tree models (Chipman et al., 2010), for example, appear as a promising technique that would also directly estimate the statistical uncertainty of gap-filled fluxes. For our estimation of

annual budgets, however, the uncertainty of statistical gap-filling is typically small compared to the systematic uncertainty of the EC flux estimates (Pirk et al., 2017).

## 5   Conclusions

We investigated snow-vegetation-atmosphere interactions at the Finse site in alpine Norway by analyzing $CO_2$ and water vapor eddy covariance flux measurements in combination with ground-based ecosystem type mapping and satellite remote sensing.

Our analysis shows the consistencies and dependencies between these fluxes, ecosystem types, and snow cover duration in three consecutive years. 2020 is identified as an extremely snow-rich year associated with a record-low SCA index for February, which reduced the total annual evapotranspiration to $50\%$ and reduced the growing season carbon assimilation to turn the ecosystem from a moderate annual carbon sink ($-31$ to $-6$ gC m$^{-2}$ yr$^{-1}$) to an even stronger source ($34$ to $20$ gC m$^{-2}$ yr$^{-1}$). As alpine tundra in Norway is less affected by disturbances such as wildfires, insect outbreaks, or heat waves, our analysis

suggests that snow cover anomalies are driving the most consequential structural shifts in this ecosystem's functioning.

*Code and data availability.*   The ecosystem type map is available on Github (https://github.com/geco-nhm/NiN_Finse). Gap-filled flux data and remote sensing results for FSCA and NDVI are available at https://doi.org/10.5281/zenodo.7566641.

*Author contributions.*   Conceptualization: NP, KA, PH, AB ; Data curation: NP, KA, YAY, AV, PH, AB ; Formal analysis: NP, KA, YAY, PH ; Funding acquisition: NP, AB, SW, TKB, FS, LMT ; Writing - original draft preparation: NP, KA, PH ; Writing – review and editing: NP,

KA, YAY, AV, ALP, AVV, PH, AB, SW, TKB, FS, LMT .

*Competing interests.*   The authors declare that they have no conflict of interest.



*Acknowledgements.* The work was supported by the Research Council of Norway (Projects #301552 "Upscaling hotspots - understanding the variability of critical land-atmosphere fluxes to strengthen climate models (Spot-on)", #294948 "Terrestrial ecosystem-climate interactions of our EMERALD planet", and #323945 "Biogeochemical processes governing boreal carbon cycling"). This work is a contribution to the strategic research initiative LATICE (Faculty of Mathematics and Natural Sciences, University of Oslo, Project #UiO/GEO103920), the Center for Biogeochemistry in the Anthropocene, as well as the Center for Computational and Data Science. The study contains Landsat data (courtesy of the U.S. Geological Survey) and modified Copernicus Sentinel data [Year 2019, 2020, 2021] obtained from the Google Earth Engine. MODIS snow-cover products MOD10A1 and MYD10A1 were obtained from NASA National Snow and Ice Data Center Distributed Active Archive Center (NSIDC DAAC).





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
