# Peer review of "Snow-vegetation-atmosphere interactions in alpine tundra"

_Biogeosciences, 2023_

## Author Comment (AC1)

**REVIEWER # 1**

COMMENT # 1.1

*The manuscript by Pirk et al. investigates links between net ecosystem CO2 and water vapour exchange and snowpack dynamics at an alpine tundra over a three-year observation period. The authors combine observations using the eddy covariance technique with remote sensing observations of snow cover and of vegetation indices and with land cover classifications. One of their main findings is that the site turned into an annual net CO2 source during the year with most snow accumulation and consequently the latest snowmelt date, while the site was a net CO2 sink during the other years when snow cover was close to the long-term mean. Understanding interactions between snowpack dynamics and land-atmosphere exchange of CO2 and water vapour is crucial to better predict climate change impacts on alpine ecosystems and the topic of this study is thus timely and addresses an important research topic.*

**Reply:**

We would like to thank the reviewer for this positive overall assessment and the thorough review that follows.

COMMENT # 1.2

*However, in my opinion, the study would strongly benefit from clearly defined research questions that better link to the authors' analyses. The submitted manuscript presents various analyses that are only loosely connected. The lack of coherence makes it difficult for the reader to grasp the most important findings/implications of this study. The authors may consider reframing the study and to link each analysis to a specific objective.*

**Reply:**

We acknowledge that the objectives of our study should be clarified to improve the coherence of the manuscript. To this end, we suggest to add the following bullet point list at the end of the Introduction section, as also suggested in Comment #1.7 below.

**Changes:**

The present study aims to explore the role of snow cover duration for ecosystem functioning in alpine tundra. Specifically, our four main objectives are to

- document the link between the presence of ecosystem types and snow cover duration for an alpine tundra site in Norway,

- quantify and determine the importance of snow cover as a driver of NEE and ET flux dynamics at the ecosystem scale,

- combine high resolution remote sensing with in-situ measurements through machine learning for flux gap-filling to quantify the annual NEE and ET balances during normal and extreme snow years,

- and contextualize the snow cover in the 2020 extreme snow year in terms of climatology using reanalysis and moderate resolution remote sensing data.

COMMENT # 1.3

*For example, it remains unclear how the spatial variability in melt-out dates across the landscape links to interannual variability in NEE and ET.*

**Reply:**

We see the observed link between melt-out dates and ecosystem types as a manifestation of the same underlying processes that create the observed interannual variability in NEE and ET, such as is sometimes employed in space-for-time substitutions. The spatial distribution of ecosystem types is mainly revealing the average conditions as it may take several decades for ecosystems to respond to changed conditions. Our flux measurements, on the other hand, resolve the fast responses and interannual variability. To clarify this view, we propose to add the following sentences to the Introduction and Conclusion

**Changes:**

**In the Introduction**

If snow-vegetation-atmosphere interactions are indeed a structuring mechanism for the ecosystem at Finse, we would expect to find responses on different temporal scales, such as (i) a large importance of snow cover variables for instantaneous flux predictions, (ii) a distinct reduction of annual NEE and ET budgets in extreme snow years, and (iii) a link between melt-out dates and the presence of ecosystem types as a reflection of the decadal average conditions. We use the eddy covariance (EC) technique...

**In the Conclusion**

Our analysis shows the consistencies and dependencies between these fluxes, ecosystem types, and the snow cover duration in three consecutive years. The spatial variability in melt-out dates of different ecosystem types (Figure 2a) and the temporal response of ecosystem-scale NEE and ET to extreme snow years (Figure 6) represent a complementary manifestation of snow-vegetation-atmosphere interactions at different

scales that determine ecosystem functioning at the Finse site.

COMMENT # 1.4

> *Furthermore, the authors demonstrate a link between snow accumulation and synoptic atmospheric circulation patterns. However, it remains unclear how this analysis then links to the flux tower measurements. The authors present a range of interesting findings that would have a much stronger impact if they were logically connected.*

**Reply:**

Our intention with the analysis of the link between snow accumulation and synoptic atmospheric circulation patterns was to provide the context for the extreme snow accumulation in 2020. We acknowledge that this analysis cannot provide a causal explanation of the snowfall dynamics, but we still think it adds important information for the rest of our study. We specifically added one item (point 4) to the bullet point list suggested in response to Comment #1.2 above that helps to motivate this analysis and connect it with the rest of the manuscript. We also propose to add a sub-sentence in the Methods part to mention that the connection between atmospheric circulation patterns and the variability in snow conditions has already been established in other studies. Finally, we propose to streamline the paragraph in Section 3.1 describing these findings.

**Changes:**

**In Materials and Methods**
Similar to the North Atlantic Oscillation Index, which has already been linked to variability in snow water equivalent in Norway (1), the Scandinavian Pattern Index is based on the surface air pressure difference between the subtropical and subpolar regions.

**In Section 3.1**
Most of the snowpack at Finse built up in only a few major precipitation events and almost half of the maximum snow depth accumulated during two snowfall events in the winter months (Figure S1 of the Supplement). The intensity of wintertime precipitation in southern Norway can in part be explained by the synoptic atmospheric circulation pattern as exemplified by the large anti-correlation between the Scandinavian Pattern Index and February precipitation shown in (Figure 3a. This correlation map shows a strong north-south gradient across Europe with large absolute correlations in many areas, indication a strong association between the SCA index and precipitation in February. ). In winter 2020, the Scandinavian Pattern Index for February exhibited its lowest value in the ERA5 record (1950-2021), while 2019 and 2021 were close to the mean value (Figure 3b). The associated large snow-fall events in the winter of 2020 contributed to the extremely late snow melt-out (Figure 3c). The beta distribution fitted to the melt-out dates  shows that 2020 falls on the $92^{nd}$-percentile of the distribution, rendering 2020 an extremely snow-rich year. The snow melt-out date in 2020 ranked $2^{nd}$ in this time series of 21 years (only exceeded in 2015).

**COMMENT # 1.5**

*In some cases, the author report correlation coefficients (r) and in some cases the coefficient of determination ($R^2$). I would recommend a consistent use of one of the two metrics.*

**Reply:**

When we want to quantify an association between different variables we use r as a measure of (linear) dependence, but when we want to quantify the predictive capabilities of a model we use $R^2$ between (nonlinear) model predictions and independent observations. In our manuscript, we only use $R^2$ to evaluate the predictive capabilities of the random forest model. We could change this to correlation coefficients, but this would seem quite unconventional. We would therefore prefer to leave these statistics unchanged.

**COMMENT # 1.6**

*Line 51-55: It remains unclear how the analysis of water-use efficiency contributes to the main goals of this study.*

**Reply:**

We agree that the analysis of WUE, showing a distinct increase in WUE in 2021, should be better rooted in our manuscript. To this end, we propose to add the following sentence to this paragraph of the Introduction, as well as in the Conclusion.

**Changes:**

**In the Introduction**
The link between the carbon and water cycles in terrestrial ecosystems can be assessed through the ratio of NEE and ET, known as the ecosystem water-use efficiency (as opposed to leaf-level water-use efficiency derived from photosynthesis and transpiration), which provides another key indicator for ecosystem functioning under changing environmental conditions (2; 3). If an ecosystem has been subject to

biochemical or biophysical shifts due to extreme conditions, one may expect to see this reflected in the ecosystem's water-use efficiency.

**In the Conclusion**

2020 is identified as an extremely snow-rich year associated with a record-low SCA index for February, which reduced the total annual evapotranspiration to 50% and reduced the growing season carbon assimilation to turn the ecosystem from a moderate annual carbon sink ($-31$ to $-6$ gC m$^{-2}$ yr$^{-1}$) to an even stronger source (34 to 20 gC m$^{-2}$ yr$^{-1}$). The ecosystem water-use efficiency increased by about 47% in the year after the extreme snow year, but longer flux monitoring is needed to assess if this response constitutes a persistent structural shift to the extremely short growing season. As alpine tundra in Norway is less affected by disturbances such as wildfires, insect outbreaks, or heat waves, our analysis suggests that snow cover anomalies are driving the most consequential  short-term responses in this ecosystem's functioning.

COMMENT # 1.7

*Line 77-84: Here, listing of the main objectives would help framing the study. As it is written now, it emphasises the "exploration" of various datasets, but I think the logical links between these analyses need to be explained.*

**Reply:**

We followed the reviewer's suggestion and added a bullet point list of objectives to this paragraph

**Changes:**

*see changes in Comment #1.2 above*

COMMENT # 1.8

*Line 92: I do not think that the "minimum 30-min average" is a good metric to support the statement that winters are mild.*

**Reply:**

We agree and instead report the winter and summer mean 30-min averages now.

**Changes:**

The climate is arctic and features maritime influences . Winters are relatively mild with December-January-February mean 2 m air temperature of 7.4°C measured between 2019-2021. Summers are relatively cool with June-July-August mean 2 m air temperature of 8.2°C measured between 2019-2021.

*Line 120: How were these limits determined?*

**Reply:**

As fluxes with larger magnitudes could potentially be more influential for the parameters of our statistical model we decided to use slightly stricter quality control thresholds for them. The chosen values for these limits of 1.0 $\mu$mol m$^{-2}$ s$^{-1}$ for NEE and 0.9 mmol m$^{-2}$ s$^{-1}$ for ET (corresponding to 40 W m$^{-2}$) were determined by visual inspection of the respective flux time series, and correspond approximately to the 40$^{th}$-percentile (after filtering) for both fluxes. We propose to mention the percentile of these limits to clarify the manuscript.

**Changes:**

We also discard data with mean horizontal wind speeds below 1.5 m s$^{-1}$, all fluxes with quality flag 2 in the scheme by (4), as well as fluxes with quality flag 1 if they have relatively large magnitudes, i.e., above 1.0 $\mu$mol m$^{-2}$ s$^{-1}$ for NEE and 0.9 mmol m$^{-2}$ s$^{-1}$ for ET (corresponding approximately to the 40$^{th}$-percentile for both fluxes after filtering).

*Line 224-225: This statement should be supported by observations. What is the contribution of February to total winter snowfall events?*

**Reply:**

This question is very difficult to answer because we are not aware of any long-term observations of snowfall magnitudes near our site, so we deliberately only mention that February features "large snowfall events". We recently installed a disdrometer at Finse to discriminate between different types of precipitation (rain, snow, sleet, freezing rain, and hail), but have not acquired sufficiently long time series to estimate robust statistics. Available snow depth measurements are complicated by other effects like snow compaction and wind redistribution, and temperaturebased precipitation phase partitioning is notoriously difficult in Norway (see e.g. http://urn.nb.no/URN:NBN:no-84028). The ERA5 total precipitation field (see Figure R1) indicates a relatively even contribution to total precipitation throughout the winter months, but does not directly represent snowfall observation. We propose a simple rephrasing of this sentence.

[Figure]

Figure R1: Average total precipitation at Finse according to ERA5 re-analysis data during the entire record of 1950-2022 (left) and during our measurement campaign 2019-2021 (right).

**Changes:**

Similar to the North Atlantic Oscillation Index, the Scandinavian Pattern Index is based on the surface air pressure difference between the subtropical and subpolar regions. We  exemplify the resulting patterns using February as a month in the middle of the snow season that typically features large snowfall events.

COMMENT # 1.11

*Figure 2: How was NDVI treated when ground was snow covered? This analysis would be strengthened if statistical analyses of differences between land cover types would be presented.*

**Reply:**

We did not apply any special processing to adjust or filter NDVI estimates for snow-covered regions or times. Fresh snow is typically characterized by low and even negative NDVI values, which are included in the presented statistics and maps, as well as in the signal used for the gap-filling model. We propose to clarify this by adding the sentence below to the Results section.

Figures 2e and 2i in our manuscript give an overview of the NDVI statistics and their differences between ecosystem types. While we agree that this analysis could be extended with additional details (e.g. through a pair-wise analysis of variance),

we believe that this would be out of scope for our study, so we would prefer to keep this analysis at the present level of detail.

**Changes:**

Averaged across the three summer months, NDVI was lower in 2020 compared to 2019 and 2021 (Figures 2e-h), which can be explained by the longer snow duration because snow-covered areas typically have low negative NDVI values close to zero.

COMMENT # 1.12

*3.2 Flux dynamics in the two footprints: The authors analyse two footprints separately, which is a reasonable approach if underlying land cover composition is very different. However, the authors find very similar NEE and ET dynamics and it remains unclear what the added value of this analysis is for the study. A better explanation of the separate treatment of the two footprints would be useful.*

**Reply:**

The drivers of NEE and ET are indeed very similar in the two footprints, as exemplified by the random forest predictor importances shown in Figures 5b and 5d. The overall magnitudes, on the other hand, are actually quite different. ET, for example, is around 90-95 mm over a normal year in footprint West, but only 60-66 mm for footprint East (see Figures 6c and 6d). Processing the flux estimates for these footprints together would yield a weighted average for the area, which would depend on the frequency of easterly and westerly wind directions. Each annual flux budget would therefore represent a mixture of spatial and temporal variability, which is not desirable in studies like ours that aim to quantify the effect of a disturbance in a particular time period (like an extreme snow year). We propose to clarify this view by adding the following sentence to the first paragraph of Section 3.2

**Changes:**

Footprint East is characterized by a larger fraction of water surfaces and late snowbeds, while footprint West has a larger fraction of fens and moderate snowbeds, with a denser vegetation cover. These two footprints are therefore treated separately to reduce the confounding effects of spatial and temporal variability of the measured fluxes.

COMMENT # 1.13

*Line 328: Perhaps rephrase: "in order to maximize leaf area"*

**Reply:**

We agree to the suggested change and propose to adjust the sentence accordingly.

**Changes:**

Further, in order to maximize leaf area, leaves that are pre-planned in buds must not be injured by low winter temperatures or frost spells after bud break (5).

COMMENT # 1.14

*Line 372: Gap-filling algorithms like Marginal Distribution Sampling (MDS) do not prescribe functional relationships.*

**Reply:**

Thanks for pointing this out. We propose to correct this sentence as follows.

**Changes:**

Unlike other commonly used gap-filling algorithms like marginal distribution sampling (6) where gaps are filled with average fluxes measured during similar conditions within a moving time window, the random forest model  implicitly models the nonlinear biogeo-chemical and -physical interactions that give rise to fluxes.

**REFERENCES**

[1] T. Skaugen, H. B. Stranden, and T. Saloranta, "Trends in snow water equivalent in Norway (1931–2009)," *Hydrology Research*, vol. 43, pp. 489–499, Aug. 2012.

[2] S. Niu, X. Xing, Z. Zhang, J. Xia, X. Zhou, B. Song, L. Li, and S. Wan, "Water-use efficiency in response to climate change: From leaf to ecosystem in a temperate steppe: WATER-USE EFFICIENCY IN RESPONSES TO CLIMATE CHANGE," *Global Change Biology*, vol. 17, pp. 1073–1082, Feb. 2011.

[3] W. H. Schlesinger, *Biogeochemistry: An Analysis of Global Change.* SanDiego: Elsevier, fourth ed., 2020.

[4] Th. Foken and B. Wichura, "Tools for quality assessment of surface-based flux measurements," *Agricultural and Forest Meteorology*, vol. 78, pp. 83–105, Jan. 1996.

[5] S. Wipf, V. Stoeckli, and P. Bebi, "Winter climate change in alpine tundra: Plant responses to changes in snow depth and snowmelt timing," *Climatic Change*, vol. 94, pp. 105–121, May 2009.

[6] M. Reichstein, E. Falge, D. Baldocchi, D. Papale, M. Aubinet, P. Berbigier, C. Bernhofer, N. Buchmann, T. Gilmanov, A. Granier, T. Grunwald, K. Havrankova, H. Ilvesniemi, D. Janous, A. Knohl, T. Laurila, A. Lohila, D. Loustau, G. Matteucci, T. Meyers, F. Miglietta, J.-M. Ourcival, J. Pumpanen, S. Rambal, E. Rotenberg, M. Sanz, J. Tenhunen, G. Seufert, F. Vaccari, T. Vesala, D. Yakir, and R. Valentini, "On the separation of net ecosystem exchange into assimilation and ecosystem respiration: Review and improved algorithm," *Global Change Biology*, vol. 11, pp. 1424–1439, Sept. 2005.

[7] N. Gorelick, M. Hancher, M. Dixon, S. Ilyushchenko, D. Thau, and R. Moore, "Google Earth Engine: Planetary-scale geospatial analysis for everyone," *Remote Sensing of Environment*, vol. 202, pp. 18–27, Dec. 2017.

[8] K. Aalstad, S. Westermann, and L. Bertino, "Evaluating satellite retrieved fractional snow-covered area at a high-Arctic site using terrestrial photography," *Remote Sensing of Environment*, vol. 239, p. 111618, Mar. 2020.

[9] G. J. Jia, H. E. Epstein, and D. A. Walker, "Greening of arctic Alaska, 1981–2001," *Geophysical Research Letters*, vol. 30, p. 2003GL018268, Oct. 2003.

[10] C. E. Rasmussen and C. K. I. Williams, *Gaussian Processes for Machine Learning.* The MIT Press, 2005.

[11] V. Salomonson and I. Appel, "Development of the Aqua MODIS NDSI fractional snow cover algorithm and validation results," *IEEE Transactions on Geoscience and Remote Sensing*, vol. 44, pp. 1747–1756, July 2006.

[12] D. K. Hall, A. Riggs G., V. Solomonson, and N. M. SIPS, "MODIS/Aqua Snow Cover Daily L3 Global 500m SIN Grid," 2015.

[13] D. K. Hall, A. Riggs G., V. Solomonson, and N. M. SIPS, "MODIS/Terra Snow Cover Daily L3 Global 500m SIN Grid," 2015.

[14] D. K. Hall, G. A. Riggs, N. E. DiGirolamo, and M. O. Román, "Evaluation of MODIS and VIIRS cloud-gap-filled snow-cover products for production of an Earth science data record," *Hydrology and Earth System Sciences*, vol. 23, pp. 5227–5241, Dec. 2019.

[15] H. A. Gleason, "The individualistic concept of the plant association," *Bulletin of the Torrey botanical club*, pp. 7–26, 1926.

[16] R. H. Whittaker, "Vegetation of the Great Smoky Mountains," *Ecological Monographs*, vol. 26, pp. 1–80, Jan. 1956.

[17] V. Vandvik, O. Skarpaas, K. Klanderud, R. J. Telford, A. H. Halbritter, and D. E. Goldberg, "Biotic rescaling reveals importance of species interactions for variation in biodiversity responses to climate change," *Proceedings of the National Academy of Sciences*, vol. 117, pp. 22858–22865, Sept. 2020.

[18] P. Niittynen, R. K. Heikkinen, J. Aalto, A. Guisan, J. Kemppinen, and M. Luoto, "Fine-scale tundra vegetation patterns are strongly related to winter thermal conditions," *Nature Climate Change*, vol. 10, pp. 1143–1148, Dec. 2020.

[19] P. Niittynen, R. K. Heikkinen, and M. Luoto, "Snow cover is a neglected driver of Arctic biodiversity loss," *Nature Climate Change*, vol. 8, pp. 997–1001, Nov. 2018.

[20] E. R. Frei and G. H. Henry, "Long-term effects of snowmelt timing and climate warming on phenology, growth, and reproductive effort of Arctic tundra plant species," *Arctic Science*, vol. 8, pp. 700–721, Sept. 2022.

---

## Author Comment (AC2)

**REVIEWER # 2**

**COMMENT # 2.1**

*Pirk and others explore the response of a Norwegian alpine tundra ecosystem to a year with anomalously late snowmelt. The Introduction bounced around a bit between different topics including carbon flux, plant succession, global change, hydrology, remote sensing, and more. All of these things are interconnected of course, and the Introduction was very nicely cited, but the topics could be linked more clearly to point toward the particular topic of this study. As a consequence it wasn't entirely clear why the landsat, sentinel, and modis observations were used when modis measures more frequently at coarser scales and landsat and sentinel measure less frequently at finer scales, and how these observations fit together. Was MODIS for historical melt out dates and how were melt out dates characterized for the 16 day landsat overpass? The results are interesting but I had a difficult time understanding how everything fit together.*

**Reply:**

We would like to thank the reviewer for the constructive and helpful comments. We acknowledge that the different topics introduced in the Introduction should be better linked to improve the logical flow in our study. We propose to implement changes as outlined in Comment #2.3 below as well as in Comment #1.2 by Reviewer 1.

Regarding remote sensing, we use merged and gap-filled Sentinel-2 and Landsat 8 imagery in the period 09.2018-09.2021 to retrieve high resolution daily FSCA and NDVI. There retrievals are used as predictors in the flux gap-filling and to analyze spatial patterns including the link to ecosystem type (Figure 2). Note that the 16-day Landsat imagery was never used in isolation, but was instead combined with Sentinel-2 imagery using Gaussian process regression resulting in merged FSCA and NDVI products. Longer-term but coarser scale MODIS imagery is used to retrieve FSCA over two decades, from which we estimate the melt-out date of the seasonal snowpack for each water year in the period 2001-2021 for the area around the Finse flux tower. These melt out dates are used to help contextualize the snow cover dynamics in the three water years 2019-2021 in terms of the longer term snow climatology around Finse (Figure 3). The quality of the MODIS data is also briefly evaluated using the higher resolution satellite imagery as a reference. This has now been clarified in the new subsection "Satellite remote sensing" as follows.

**Changes:**

[revised manuscript text omitted]

**COMMENT # 2.2**

*Are fig. 5 c and d on log scales?*

**Reply:**

Yes indeed, the y-axes in these plots are scaled logarithmically. We have clarified this by adding more y-labels to these axes in the revised version of the figure (see Figure R2), and additionally noted this in the figure caption.

**COMMENT # 2.3**

*The eddy covariance measurements were discussed nicely and the gapfilling approach was well suited to the site. All in all with some restructuring and focus on a consistent narrative the manuscript will be publishable as it makes some interesting points.*

**Reply:**

Thanks for this positive evaluation.

We also worked more on the structure of the manuscript. The method subsection on "Satellite remote sensing and synoptic patterns" has now been split into two subsections. The resulting subsection on "Satellite remote sensing" was moved before the subsection on "Flux gap-filling" because two remotely sensed variables are used there. We also added more references between the subsections of our methods to improve the coherence of the manuscript.

[Figure]

Figure R2: Flux dynamics and drivers. Left: Gap-filled NEE and ET as fingerprint plots for footprint West. Right: Predictor importance of the random forest regression models of both footprint East and West plotted on a logarithmic scale. Black error bars indicate the standard deviation across the 2000 trees in the respective random forests.

To improve the logical flow in the Introduction, we propose to implement the following changes, aiming to highlight the specific topic at the beginning of a few of the paragraphs.

**Changes:**

Community ecologists have long recognized that plant associations form and thrive in specific ranges of environmental conditions (15; 16).  However, snow-vegetation interactions and the related responses to snow cover changes in high latitude and altitude ecosystems can be highly context-dependent (19; 18). (5) and (20) analyzed...

...

 There are a number of indicators for an ecosystem's interaction with the atmosphere that—while related—highlight different aspects of this coupling. The Normalized Difference Vegetation Index (NDVI), for example, has been used to document widespread greening of mountain slopes (9). Such changes can in turn have profound impacts on the ecosystem's carbon and water balances...